# Developing an applied model for making decisions towards the end of life about care for someone with dementia

Nathan Davies[1,2]*, Tanisha De Souza[1], Greta Rait[1], Jessica Meehan[1], Elizabeth L. Sampson[2,3]

1 Centre for Ageing Population Studies, Research Department of Primary Care and Population Health, University College London, London, United Kingdom, 2 Centre for Dementia Palliative Care Research, Division of Psychiatry, Marie Curie Palliative Care Research Department, University College London, London, United Kingdom, 3 Barnet Enfield and Haringey Mental Health Trust Liaison Team, North Middlesex University Hospital, London, United Kingdom

* n.m.davies@ucl.ac.uk

## Abstract

### Background

Many people with dementia reach the end-of-life without an advance care plan. Many are not ready to have conversations about end-of-life, and decision-making is left to their families and professionals when they no longer have capacity. Carers may benefit from further support with decision-making. To develop this support, it is important to understand the decision-making process.

### Aim

Explore with family carers and people living with dementia the decision-making process and factors that influence decision-making in dementia end of life care, to produce a model of decision-making in the context of dementia end-of-life care.

### Methods

Semi-structured interviews with 21 family carers and 11 people with dementia in England (2018–2019) from memory clinics, general practice and carer organisations. Interviews were analysed using thematic analysis and findings were mapped onto the Interprofessional Shared Decision Making model, refined to produce a modified model of decision-making in dementia.

### Results

Participants described five key decisions towards the end-of-life as examples of decision making. We used these experiences to produce a modified model of decision-making in dementia end-of-life-care. The model considers the contextual factors that influence the decision-making process, including: personal preferences; advance care planning and Lasting Power of Attorney; capacity and health and wellbeing of the person with dementia; support from others and clarity of roles. The decision-making process consists of seven inter-

**Data Availability Statement:** All relevant data are within the paper.

**Funding:** This work was supported by Alzhiemer's Society [grant number: AS-JF-16b-012]. ND, GR,

ELS received the award. The funders had no role in study design, data collection and analysis, decision to publish, or preparation of the manuscript. https://www.alzheimers.org.uk/.

**Competing interests:** The authors have declared that no competing interests exist.

linked stages: 1) identifying the decision maker or team; 2) sharing and exchanging information; 3) clarifying values and preferences; 4) managing and considering emotions; 5) considering the feasibility of options; 6) balancing preferred choice and the actual choice; and 7) implementation and reflecting on outcomes.

## Conclusions

The modified model breaks down the decision-making process and attempts to simplify the process while capturing the subtle nuances of decision making. It provides a framework for conversations and supporting decisions by carers.

## Introduction

People living with dementia have a right to be involved in decisions about their care, many countries have passed laws to this affect, however this may be more prominent in developed countries. However, dementia affects an individual's ability to retain information, weigh up options and make judgements, ultimately diminishing their decision making capacity [1]. Many with dementia will increasingly require the support of those around them, including family or professionals to make decisions. This may include decisions about healthcare and treatment, living arrangements, finances and daily activities of living. In the later stages of dementia and towards the end of life the person living with dementia is unlikely to have capacity to make their own decisions.

When the person can no longer make their own decisions they are made by a surrogate decision maker or proxy, this is usually a family carer or a professional depending on the significance of the decision to be made and national or local legislation. For example, in the UK the Mental Capacity Act states for those who lack the capacity decisions should be made in their best interests [2]. In addition, some people with dementia in the UK have a Lasting Power of Attorney for Health and Welfare, made out when they still had capacity, which enables carers to make these decisions. The terminology and legality of such procedures will differ internationally, for example in the UK Lasting Power of Attorney needs to be registered with the Office of the Public Guardian for it to be legal.

There have been many international initiatives to encourage advance care planning with people living with dementia early, to plan for their care including their end-of-life care (EOLC) [3, 4], making choices for when they no longer have capcity. However, people report a lack of conversations about EOLC and planning for the later stages of dementia [5], and often reach the end-of-life without a previous discussion about their choices for care [6]. End of life discussions may include where someone would like to be cared for and die, what are their preferences for care and treatment, what treatment they would not like to receive, and importantly what care they would like to receive. End of life is difficult to define in dementia due to the uncertain nature of the condition. The dementia trajectory is variable characterised by progressive decline, and punctuated by acute events such as an infection or falls, where an individual may recover or experience an increased rate of decline in health until their death [7].

Many people with dementia and family carers may not want to have discussions about EOLC [8]. Additionally, some practitioners lack the confidence to hold EOLC conversations, overlook the importance of ongoing discussions about care planning, or simply do not have the time [9, 10].

Studies have demonstrated that family carers find making decisions difficult and complex, managing the stress of making decisions and challenging professionals with sometimes minimal support from those around them [11]. Carers have described their role as a 'patient manager' and found EOLC decisions particularly difficult, mainly medical decisions about treatment and resuscitation [12]. However, many carers were helped when they knew the preferences of the person and were supported by those around them [12].

Carers may benefit from further support with decision making for example from simplified frameworks, decision support or decision aids [13–15]. To develop this support, it is important to understand the decision-making process and important factors influencing decisions from the perspective of people living with dementia and family carers. Relatively little work has included the views of people living with dementia about EOLC [16, 17]. The majority of work on decision making in dementia has focussed on the barriers and experiences of making decisions on behalf of someone else and little has been done to understand and conceptualise in depth the decision-making process.

Models of decision-making have been developed in the field of shared decision-making. The Interprofessional Shared Decision Making (IP-SDM) model is one such example, breaking down decision making to consider: the context, the role of those involved and the steps in the decision-making process [18]. The model helps to break down the complexity of decision-making and clarify where and what support is needed for decision makers. IP-SDM has been used to understand decisions made by caregivers of older adults [19], experiences of people living with dementia, family carers, and professionals making decisions about housing transitions [20], decisions about eating and drinking [21]. No studies have considered the use of this model to break down the complexity of making decisions about dementia EOLC.

### Aims

We aimed to:

1. Explore the process and factors influencing decision making towards the end of life from the perspective of family carers of someone living with dementia.

2. Explore the perspectives of people living with dementia on family carer decision making towards the end of life.

3. Map the process and factors influencing decision making to the IP-SDM model

4. Refine the IP-SDM model in the context of dementia EOLC to produce a refined model of decision making.

## Methods

### Ethics

Ethical approval was obtained London Queen Square Research Ethics Committee (18/LO/0408) Written informed consent was obtained from all participants.

### Design

A qualitative study using semi-structured interview methods analysed using thematic analysis methods and the IP-SDM model to develop a model of decision making in the context of dementia towards the end of life. Throughout this study we were advised and guided by our patient and public involvement group which consisted of four former family of carers of people living with dementia, whose role was to apply their expertise from their lived experiences.

## Participants and recruitment

People living with dementia and family carers of people with dementia were purposively sampled. Participants were recruited from a number of sources including memory services, general practices, and national dementia/carer networks. Eligibility criteria (see below) were provided to sites and eligible participants were identified and invited by clinical staff in writing or in person. Participants were also recruited from national dementia/carer networks, including the National Institute for Health Research Join Dementia Research website. Invitations were sent by members of the research team or the host organisation via email. Interested participants contacted the research team directly.

## Inclusion and exclusion criteria

Participants were included if they were:

- Family member or friend who provided unpaid care for a person living with dementia in the later stages of dementia or towards the end of life

- Current or former carer

- Surrogate decision maker for the person living with dementia, either informally or through Lasting Power of Attorney for health and welfare

- Person with a clinical diagnosis of dementia

- Able to provide informed consent

- Read and speak English

    Participants were excluded if:

- They experienced a bereavement in the last three months to minimise distress to participants as per previous studies [22] and advise from our Patient and Public Involvement (PPI) group

- There were clinical or social concerns that precluded them being approached

   We ensured participants had the capacity to consent by conducting a short capacity assessment before we began the interview, using the principles of the UK Mental Capacity Act [2]. This required us to check they understood the study through a brief discussion and whether they could retain, use or weigh the information as part of making the decision and communicate their decision.

## Procedure

Following invitation participants provided written informed consent at the start of the interview. Semi-structured interviews took place at the participant's home or researcher's university guided by a topic guide developed from the literature [5, 14, 23]. We used vignettes to prompt thinking about the process of making decisions and factors influencing decisions [24]. The topic guide and vignettes were pilot tested with two participants initially, which were included in the final analysis. Minor amendments were made to the topic guide for phrasing based on pilot interviews and iteratively refined throughout the study. Interviews were conducted individually or in dyads with the person with dementia and a family member. In dyads we were interested in hearing the views of both participants. Interviews were conducted by either ND or TDS both researchers are experienced in qualitative research with backgrounds in psychology and dementia care, they also led on analysis. A distress protocol was developed if

participants became distressed during the interviews, including pausing or stopping the interview, providing contacts for support and contacting the participants by telephone after the interview.

## Conceptual model

In this study we based our interpretation of what constitutes a model from Nilsen (2015) who asserts a model deliberately simplifies a phenomenon, and need not always be an accurate representation of reality to have value. They have been compared to theories but theories are explanatory and descriptive, however a model is more descriptive than explanatory [25].

The IP-SDM was used as a conceptual model to guide and shape the organisation, presentation of findings and modified to develop a refined model of decision making in dementia EOLC [18]. This model was chosen as it was developed for decisions in primary care which delivers most clinical dementia care. It emphasises decision making as a shared process which is emphasised in dementia care and considers factors at the macro, meso and micro level. The model provides a stepwise framework to break down the process, consisting of the individual steps of the decision-making process, including: identifying which decision to be made, exchange of information, eliciting values and preferences, considering the feasibility of the options, the preferred choice, the actual choice, implementing the decision, and outcomes (e.g. consequences) of decisions made. The model considers the context in which decisions are made, considering the social norms, organisational routines, and institutional structures. The model also considers the people involved and their roles.

However, this model was not developed considering people with dementia, many of whom lack capacity and are not able to make decisions towards the end of life. Dementia often progresses slowly and there are multiple points both within the health and social care of the individual which require decisions which this model has not considered during development. The IP-SDM assumes a linear process to decision making however, previous research has shown decision making in dementia and in dementia end of life care is very complex and may not fit a linear process [26, 27]. The IP-SDM model has been developed focussing on inter-professional decision making, however the current study is exploring decisions by families and people living with dementia themselves. Finally, developed in primary care this model has several aspects which focus on system level factors which may need modification when decisions are being made by family carers.

## Analysis

Interviews were audio recorded, transcribed verbatim, anonymised and imported into NVivo 12 software. The data was analysed using thematic analysis methods from the start of data collection as an iterative process [28]. Three researchers (blinded for review) independently coded three transcripts and researchers met to compare coding and devised a coding list. The coding list was then applied to the remaining interviews by two researchers. Coding was discussed and revised in regular meetings among researchers this also helped to confirm when we had reached data saturation. Data was managed using NVivo software.

Following an iterative process codes were mapped onto the IP-SDM stages, through regular discussions among the whole research team. We discussed the fit of codes and searched for cases and examples which did not align with the stages of the IP-SDM. This allowed us to identify gaps and stages missing from the IP-SDM model and critique the applicability in this field. Through these discussions we identified common patterns among the codes to construct themes, guided by the IP-SDM as well as remaining grounded in the data. This allowed us to refine the model by renaming, redefining, and adding new stages to fit decision making in this

population. Our themes consist of the stages of our refined decision-making model and sub-themes highlight the factors to consider within the decision-making stages. We presented the model and supporting data to our PPI group to increase rigour of our model and findings by generating discussion and ideas [29]. They provided feedback on how the model resonated with their experiences. This follows similar procedures in other studies [20, 21].

## Results

Interviews were conducted with 21 family carers and 11 people living with dementia (including six dyads). In the dyads it varied as to who led most of the discussion. Participants predominantly identified as white British (64% of people with dementia, 57%). Family carers were mainly still caring (76%), female (81%), and ranged from 41 to 86 years of age. People with dementia ranged from 72 to 90 years of age, and over half (55%) were male.

We present the decisions which participants identified, the contextual factors influencing the decision-making process and finally, the stages of the decision-making process with factors to consider within these stages. The themes below are the key aspects of the refined IP-SDM model which is presented in Fig 1.

### Decisions

Family carers discussed five overarching key areas of decision making towards the end-of-life when discussing the process and factors influencing decisions: 1) ensuring everyday wellbeing of the person with dementia; 2) managing eating and drinking difficulties; 3) managing changes in care, including transitions in place of care; 4) managing healthcare and treatment; 5) managing financial and household affairs.

### Context

We identified four contextual factors which influenced the process of making decisions by carers: 1) personal preferences; 2) advance care planning and Lasting Power of Attorney; 3) capacity, health and wellbeing of the person with dementia; 4) support from others and clarity of roles.

**Personal preferences.**  Openness and willingness to engage in discussions varied. Adult children carers attributed this to a generational difference. They felt their parents did not want to discuss death and dying, but as a younger carer they were much more pragmatic and had thought about what may happen in the future:

*'I'm very much of that mindset, like, "Okay, we need to talk about death and I want to live out the end-of-my life in an appropriate way," whereas they [parents] would be probably in a slightly–I think it's also a generational thing of just like a naivety about it. You don't talk about death, it's all kind of brushed under the carpet' (Carer, C016)*

Family carers appeared to want to place a reason on why people were not willing to discuss their EOLC wishes, generation may be just one factor, but often it simply appeared to be down to personal preferences. An individual's outlook on life and personality influenced discussions and decision-making, for example not wanting to tempt fate.

**Advance care planning and Lasting Power of Attorney.**  Previous discussions appeared to focus broadly on finances; what happens when someone dies, in particular funeral planning, and hence more general planning. There was less advance care planning about end-of-life, with a few only mentioning do not attempt cardiopulmonary resuscitation (DNACPR), preferences for care and treatment or Lasting Power of Attorney for Health and Wellbeing. This

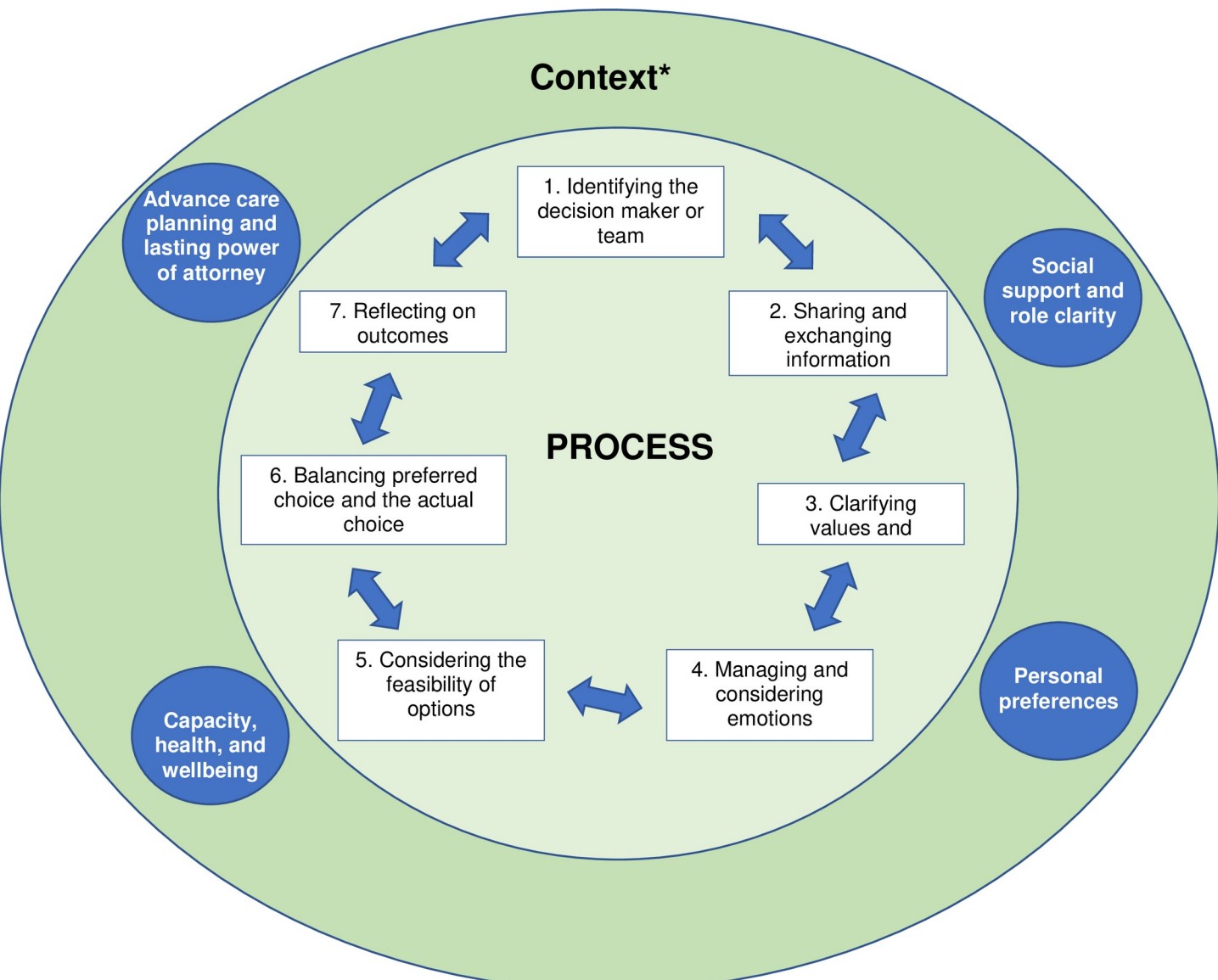

**\* Throughout the process of making decisions the following questions relating to the context need to be considered:**

- What are the individual's personal preferences?

- Is there an advance care plan?

- Is there a Lasting Power of Attorney?

- Is there clarity of roles of those involved in the persons care
  and decisions about their
  care?

- Does the person have capacity?

- How is the person's health, and wellbeing?

- What support is there from others?

**Fig 1. Conceptual model of decision-making in dementia EOLC.**

resulted in a lack of preparation to support family carers in making decisions later and also the legal power to make decisions. When medical decisions were discussed in interviews both family carers and people living with dementia appeared to not fully understand what the decisions meant and how they would help them in the future:

> Interviewer: 'Have you had any discussions with anybody about the future and what you might want in the future?
>
> Participant: No, well, let's just say, I said when I die, I want to go in a bluebell wood' (Person with dementia, P037)
>
> Interviewer: 'Why is that do you think [you have not discussed end-of-life]?
>
> Participant: I don't know. Are we supposed to have one?
>
> Interviewer: Do you think you should?
>
> Participant: Perhaps I–the GP has signed a CPR [cardiopulmonary resuscitation]? Or CRP? They have signed that for me and I've signed it.' (Wife of person with dementia, C013)

**Capacity, health and well-being of the person living with dementia.**   Perceptions about physical health, psychological wellbeing, and capacity, affected when and how decisions were made including options available, for example moving into a care home. People with dementia discussed their capacity in terms of their level of awareness. They thought this was a high priority to consider when making decisions, not just about when a carer or someone else should make a decision on their behalf, but they also associated this with their physical decline and abilities as a sign to increase support:

> 'At what point do I give up living where I am? I think it's when I become bedridden, that's one. Two–when I don't know what I'm doing. And three–if I'm in a lot of discomfort and pain or something like that.' (Person with dementia, P039)

For many family carers, they were caring for someone who had lost capacity, however for some they highlighted the need to not simply dismiss the voice of the person with dementia and to consider carefully their values, and their level of capacity to input into decisions:

> 'He's not a child. He has a right to make choices, they might be the wrong choices, but from my perspective.' (Carer, C017)

**Support from others and clarity of roles.**   Many family carers in particular, talked of conflicts in their relationships with friends, neighbours and family members. Often the societal stigma associated with dementia also contributed to carers' stress and created difficult environments for caring and decision-making, sometimes forcing decisions:

> 'That was the trigger point for the talk of a care home. It's actually a couple of run ins with neighbours, I would say, that feeling that the neighbourhood is starting to judge.' (Carer, C007)

A lack of coherence, unclear roles and need for negotiations within families created difficulties in making decisions. Many family carers discussed conflict around workload, finances or differences in the emotional responses to the outcome of some decisions. Not having strong relationships initially and living apart from one another, fueled disagreement. Living apart

meant that disagreements were not always able to be dealt with face-to-face but sometimes took place over phone messages, heightening tensions:

*'It [Family WhatsApp group] became about who's not doing stuff and kind of like a platform for people to vent frustrations. And my youngest sister and my oldest sister have a lot of rows on it, to the point–I was actually on holiday when it all kicked off. And my older sister, has now left the group.' (Carer, C007)*

## Process

The process of making decisions consisted of seven stages: 1) identifying the decision maker or team; 2) sharing and exchanging information; 3) clarifying values and preferences; 4) managing and considering emotions 5) considering the feasibility of options; 6) balancing the preferred choice and the actual choice; 7) implementation and reflecting on outcomes.

**Stage 1: Identifying the decision maker or team.**   The core role of the key decision maker was to advocate for the individual. These decisions often began with financial decisions, and decisions about daily life. As the dementia progressed so did the responsibility associated with decisions and the role became more focused on making decisions about care, with a heavier emotional burden. Whether making decisions alone or with others, decisions were never easy to make, this was sometimes reflected when carers refused to formalise plans:

**'Why was your dad [caring for his wife] so adamant not to have the DNR?**

*I think it's just because he felt it was almost like, you know, life support button, who's going to switch it off?' (Carer, C016)*

Many adult children faced an internal dilemma when making decisions, what was their rightful role and place? Despite their age, they still viewed themselves as the child. In many cases adult children did not want to make decisions and wanted someone else to make these or have 'permission' from other family members:

*'We almost still are children. You still stay in that role, in a way, when you're grown up and you've got your own lives. So almost we needed dad to say it's okay. [. . .] even though you're losing your mum, she's still your mum. And you're still her child.' (Carer, C007)*

Many people with dementia were unable to think about their future health in particular how their health might deteriorate to the point when they would no longer have capacity. Those who were able to express feelings about EOLC also valued their continued involvement. Carers often continued to include the person with dementia for as long as possible and cautioned that the current view expressed by the individual should not be ignored.

People with dementia were particularly distressed with the idea that family members may have to make decisions which may involve making the difficult decision to stop treatment:

*'Although, you see, I can't ask them to make a decision to let me go [. . .]. I hope I'd be so semi-conscious that [. . .]. Because you're imposing on somebody else, terrible decision.' (Person with dementia, P026)*

Some people with dementia differentiated the identity of the decision maker by the nature of the decisions. Decisions around place of care and supporting the person on a day-to-day basis were seen as more acceptable for families to make, as they did not appear to have the

same emotional consequences as some clinical decisions. However, clinical decisions with an end-of-life focus were to be made by professionals, but still including family, so as not to negatively impact on the emotional wellbeing of the family. This appeared to be a mechanism of relieving burden and guilt families may feel:

*'[If] I can't make decision [about a natural death], I think, in that situation, of course, I'm not expecting anybody to make the decision for me. I will say let doctor make the decision. (Person with dementia, P012)*

**Stage 2: Sharing and exchanging information.** There is a need to share and exchange information about care and the health of the individuals through the decision making process. When sharing and exchanging information, there should be a consideration that many may be in fear or in denial, and avoid discussions, for example people with dementia may fear losing agency.

*Lack of information and discussions*. A lack of detailed discussions about dementia and planning EOLC meant families were often ill informed about an individual's wishes for EOLC and hence not prepared. The earlier themes, highlighted the reluctance by some people with dementia to engage or instigate conversations, but professionals sometimes acted as a barrier to information exchange by appearing to not instigate conversations. This was important as families often relied on professionals to initiate conversations:

*'We were called in to see the GP in the nursing home. And I thought we were going to have a DNAR conversation, and I thought, yes, completely appropriate. [. . .] But on this event, [. . .] the doctor came, she didn't raise it. And I was like, okay. . .'* (Carer, C014)

The lack of planning and clear communication between all involved resulted in many family carers feeling lost, alone and out of control, with multiple descriptions of decision-making in the moment of a crisis:

*'It's crisis management a lot of the time'* (Carer, C009)

*Questioning applicability to 'us' as an expression of denial, fear and avoidance.* Several family carers talked of their discomfort at the idea of discussing the future and preferred to face problems or difficulties as they arose. Many people with dementia and family carers were avoidant of discussing the future and either explicitly or implicitly expressed denial about a physical and cognitive decline, prospect of dying or may happen at the end of life; leading to a lack of information exchange. In some occasions this stemmed from the beginning of a diagnosis and denial and disbelief of their diagnosis:

**Interviewer: *'Have you had those kind of discussions [end-of-life] with them in the past?***

*Participant: I don't think so, no. Me? I'm never going to die, I'm fit! No, I haven't discussed it with them. I suppose that's a form of cowardice, isn't it?*

**Interviewer: *Do you think it's helpful to have those kind of talks?***

*Participant: Once. Not to keep on. On about it. As long as, as long as they know that I wouldn't object. That's enough. There's no need to keep going back and talking about it.'* (Person with dementia, P026)

*'I think also we're [family] massively in denial about what happens at the end. So my younger sister, who's the nurse says, you know, mum won't be able to eat, she won't be able to do this– and I'm like, well, no that didn't happen to her dad. So we've got this reference point of our granddad. And I'm like, "No, no, he just went to hospital and died, I think, of pneumonia or something." So I think we're, I think we're massively in denial.'* (Carer, C007)

*'The psychologist has explained to me that my father's attitude towards his diagnosis is denial. And he, he thinks and believes that he doesn't have a problem, he never will have a problem'* (Carer, C007)

This reluctance to engage in discussions, often resulted in a lack of exchange of information and the assumption that conversations were not needed as the family would know what to do if and when the moment arose:

*'I just think I'm sure my children will sort it out if they have to'* (Person with dementia, P037)

*Loss of agency and control.* For some people with dementia they appeared to avoid discussions out of fear of losing agency and relinquishing control. Discussions were seen as a way of acknowledging and confronting these fears or handing over control, as represented in the dyad below:

**Interviewer: 'What do you think about Power of Attorney?**

*Person with dementia: Yes, yes we can do it, yes, after this conversation probably I should seriously consider it. My daughter reminds us many times we should have a will.*

**Interviewer: And why is it that you've not done that?**

*Carer: [he says] It's not right, it's not yet, you see. But it's time now [. . .]*

*Person with dementia: I don't think that's that desperate.*

*Carer: I don't think, I don't think he wants to give up his power. Normally, he's quite dominant, you see.*

*Person with dementia: No, no, not yet. (Person with dementia, P012; and Carer, C018)*

A second family interviewed also suggested the difficulty the person with dementia had acknowledging his loss of power, and control which represented his agency:

*'No, this is the first time that it's–we haven't spoke about [the future]–because I didn't think that I was about to pop my clogs.'* (Person with dementia, P023)

The person with dementia continued throughout the interview to affirm he had always been head of family, making all the decisions, and did not see why that would change. The loss of control and agency appeared to symbolize a change in status for some from a provider to a dependent, and a position of strength to a position of weakness:

*'I've always been a leader'* (Person with dementia, P023)

**Stage 3: Clarifying values and preferences.** It is important to acknowledge that when making decisions on behalf of others there may be multiple decision makers that could

increase difficulties and it is important to clarify the values of all involved, which may help manage any disagreement or conflict. Family carers discussed the presence of or lack of a shared understanding, goals, values and wishes amongst decision makers. Differences in views was often a result of increasing care needs and discussions about where the person with dementia should live:

*'I think he [brother] doesn't understand the emotional involvement. [. . .] we have discussed it with varying levels of success. [. . .] I think, at first, I was terribly frustrated and I kind of–I suppose I was kind of–it was like a cry for help. It's like, "For God's sake, can we do something?" And his answer was always, "Well she's going to have to go into a home." ' (Carer, C004)*

For some family carers disagreement manifested as conflict, blame and family members disengaging from caring and the wider family:

*'She [step-daughter] wouldn't dream of discussing. Or, if you tried it, she would just tell you to–"You're going to try to get rid of him [person with dementia], you know, you're going to kill him off." You know, that would be her response.' (Carer, C024)*

Knowing the right thing to do was a balancing act, filled with uncertainty. The main decision maker often had to balance all these different views and wishes, with their own views of the situation, and those of the person with dementia themselves, if known:

*'There is a judgement there that you have to ask yourself–"When I'm saying what is right, is it what is genuinely right for her or is it right for me?" And those are sometimes competing priorities. I like to think that they are more or less the same. But there are times when they might not be.' (Carer, C001)*

**Stage 4: Managing and considering emotions.** Many family carers feared upsetting the person with dementia, with discussions or making decisions. Family carers often had to judge the moment and read between the lines about when they could have discussions:

*'We haven't looked at it [information about place of care including hospice and care homes] since they [hospital] gave it to us in the sort of spring or something, you know, because you could tell [when he had the information], he didn't like [. . .]. You could tell what he meant was he didn't really like to talk about that. That's what I sort of interpreted it anyway, you know.' (Carer, C025)*

In addition to the person with dementia's emotions, family carers described how they had to manage their own emotions and often leave their emotions out of the decision-making process and consider the person's best interest:

*'The question was [after making a decision to stop treatment], 'Can I live with this is she dies?' And I thought, "Yes, I absolutely can. But can J [stepfather] live with this if she dies?" And so I was left with that. And I'm quite angry that I was left with that. But not so much that I would have sort of changed it.' (Carer, C014)*

**Stage 5: Considering the feasibility of options.** The feasibility of decisions appeared to revolve around two main strands: 1) finances; and 2) impact on the family. Woven through both these considerations was the notion of prognostic uncertainty.

*Finances*. Prognostic uncertainty created difficulty for planning the delivery of care. Family carers had to consider how affordable a change to the provision of care was. This was difficult when considering the potential long term and uncertain nature of the condition, and that many were not eligible for free care:

*'But obviously you don't get a life expectancy, do you? You know, you sort of, her dad had Alzheimer's and he died at 84, 85. So we're kind of like, "Jesus, another ten years." So if you've got ten years and you've got to financially plan for that, then. . .' (Carer, C007)*

*Impact on the family*. People with dementia emphasised family carers should consider the emotional and physical impact a decision would have on the whole family. People with dementia who accepted the decline of their dementia were mindful and concerned of the *'burden'* they may place on their families:

*'If I think I was really a burden to somebody, you know, then it would be better to say, "No I don't want that, I don't want that." And then go.' (Person with dementia, P036)*

Although for those that were unable to accept the decline of their health, they were less aware of the impact this may have on family, similarly they were less inclined to share the decision making with family. As illustrated in the quote below, they focused on the present, considering what they could do now as opposed to thinking about the future, support they may need and the impact on the family:

*Participant: 'I suppose eventually you sort of run out of brain power and you have to be looked after or pass away gently in a corner somewhere.*

Interviewer: ***And when you say 'looked after', what do you think that means?***

*Participant: Well I can wash up, I can, I haven't cooked very much since we've been married.'* (*Person with dementia*, P023)

Some family carers were clear that as the dementia progressed it became harder to continue to support the person. Care at home had an impact on the carers physical and emotional wellbeing, as well as the relationship with the person. Carers often described a *tipping point* where the current situation was no longer feasible:

*' when I cannot, when [husband] himself, with my minimal supervision, cannot manage his basic dressing, teeth brushing, going to the toilet–at that point I'm, I feel like checking out, because I can't do that. I cannot do that. I'm not there to be a–that kind of carer. [. . .] It's not what my life will be.' (Carer, C005)*

**Stage 6: Balancing the preferred choice and the actual choice.** Family carers often described the ideal situation and options. For many family carers the preferred option and choice was what was best for the person and what they would have wanted. However, the feasibility as discussed above often meant that the preferred option differed from the actual choice made. The discussion of the reactive and *'crisis management'* nature of decisions also demonstrated how actual choices were made, differed from the ideal. One family highlighted this in their decision process of picking care workers to provide support at home:

*'As a family, [we] sort of do this thing where we–"Oh my cleaning lady has found someone."
"Yes, yes, can she just come?" "Yes, yes, just send her." "So and so knows so and so." Like we
don't do much vetting, is what I'm saying. It's just desperate.' (Carer, C007)*

**Stage 7: Implementation and reflecting on outcomes.** Family carers discussed through-
out the interviews the impact on them as the decision maker, and reflected on the wider impli-
cation on the whole family of the decision making process, as discussed in the feasibility
section above. Outcomes discussed were predominantly negative with few seeing positive out-
comes of the decisions they had to make. Outcomes were discussed in relation to the carer's
physical health, in addition to their emotional and psychological wellbeing, including feelings
of grief, guilt and distress:

*'I lost income, I lost status, I lost savings, I lost pension. And all of that made me feel very
depressed and desperate' (Carer, C001)*

Family carers also discussed how the outcome of some of the decisions they made also
affected their relationships and social life, this reflects the cyclical nature of decision-making as
this also feeds back into the context in which decisions are made:

*Participant: 'We had to move her into the living room, you know, because going upstairs–the
bedroom upstairs that she was in, was very, very, very tiny. And even if we swapped rooms
with one of the larger double bedrooms, it still wasn't big enough to take all the equipment.*

***Interviewer: Okay and so what impact did that have by having her downstairs in the
lounge?***

*Participant: Yes, for me, not such a big deal. I was just quite accepting of it. [. . .] It did on the
kids.*

***Interviewer: Okay, in what way?***

*Participant: Well, they were too embarrassed to bring friends around.' (Carer, C019)*

Importantly decision-making was not a linear process, instead it was a continuous and iter-
ative process, with decisions revisited and updated throughout the course of providing care.

## Discussion

This is the first study to our knowledge to explore a decision-making model for dementia
EOLC and revise and adapt the IP-SDM model to this situation, producing an applied model.
This is a theoretical development which can be used to underpin future interventions in this
field with practical applicability to clinical practice.

### Revised model of decision making

The IP-SDM model has previously been revised in other studies to fit different contexts and
populations of interest which the model was not originally developed for [20, 21]. We followed
a similar approach to these studies and revised the model for the dementia end of life care con-
text through a thematic analysis.

To develop a model of decision making in the context of dementia EOLC and understand
the decision making in more detail we mapped our findings on to IP-SDM conceptual model

[18], which acted as a theoretical foundation. The main components and stages in our revised model reflect the main principles of the IP-SDM as outlined in the methods, however we have revised the model to fit this population (see Fig 1). Our revised model details the stages of the decision-making process, highlights factors which need to be considered within the stages and the context in which decisions are made wraps around the decision-making process.

**Context.** The original IP-SDM model presents a top-down influence of the environment on decision making which includes the social norms, organisational routines, and institutional structure. The model is focussed on inter-professional working and was developed for the primary care setting and has a large emphasis on professionals. However, our model focusses on the decisions made by families and with people living with dementia and not specifically within primary care. Our revised model reconceptualises the environment as contextual factors, evidenced in our findings. In our model contextual factors influence the decision-making process and have a greater emphasis on the individuals as opposed to the organisational structures and routines, which were not discussed by participants. In our revised model contextual factors wrap around the entire decision-making process as can be seen in Fig 1, and may differ depending on the type of decision which needs to be made. Previous discussions including those regarding advance care planning, the health and well-being of the individual and their personal preferences will all influence the decisions being made. Although we have placed advance care planning and previous discussions in the context in our model, they may also be an outcome of the decision-making process. This once again highlights the cyclical nature of decision making and how the process and context interact and influence one another.

Our findings demonstrate the level of social support, including societal influences and relationships among decision makers influence the decision-making. We suggest these societal and social factors are more prominent in dementia as there are different home and care contexts, as well as heightened societal stigma [30], and have greater importance in our revised model as demonstrated in Fig 1 as its own component in the context.

**Decision making process.** The original IP-SDM model presents a series of linear steps of the decision-making process. However, this does not reflect the complexity as reflected in our results. We have revised the model to reflect the cyclical nature of decision-making (see Fig 1), as described in other refinements of this model [21]. The six stages within our model are not independent, they are inter-linked and influence one another in a cyclical manner. This is more emphasised in dementia decision making where proxies are making decisions, and not the patient themselves as described in the IP-SDM model. This highlights the need for a revised model adapted for the context of dementia EOLC. Making decisions as proxy is marked with uncertainty at each stage [26], which forces people to fluctuate between stages, and revisit decisions. Within each of the stages we have presented key factors to consider by the decision maker and anyone supporting the decision maker.

It is important the individual or team making decisions are identified and clarified early. Importantly our study highlights that many people with dementia still wanted to be involved in decision making as much as possible which resonates with other studies [31]. However, previous research on decision making has highlighted people with dementia are often excluded because of assumptions they have a lack of capacity [32]. When they lost the ability to remain involved or be the decision maker, ultimately people with dementia generally wanted their families to make decisions. However, although everyday decisions about care and wellbeing were expected to be made by families, decisions which were focussed on more medical aspects of care and in particular care about end of life, professionals were thought best to make these decisions and consult with families. This is crucial for families to understand as many feel responsible and a large burden for making decisions as proxy [33, 34], they should be reassured this is not always the case, especially for emotionally laden medical based decisions.

When sharing and exchanging information, our findings show there should be a consideration that people with dementia and some family carers may be in denial, and avoid discussions, for example people with dementia may fear losing agency. It is important to acknowledge that when making decisions on behalf of others there may be multiple decision makers that could increase difficulties and it is important clarify the values of all involved, which may help manage any disagreement or conflict. As in previous studies many participants did not seem prepared for end of life decisions and had only considered topics such as arrangements for after death as opposed to care before death [34, 35].

A novel element to our refined model is the addition of a stage 'managing and considering emotions' which was missing from the original IP-SDM model. From our results this is crucial in dementia, with decisions made as proxy, carers may be making decisions at times when they are experiencing high levels of anticipatory grief [36]. It is therefore important to consider the emotional impact of these decisions as they will feed into the longer-term grief process [35].

The feasibility of the options is considered in the IP-SDM however, for many participants the feasibility in dementia was focused on finances and the impact on families who provide the majority of care for people living with dementia in the UK [37]. As this study was conducted in the UK many options around place of care for example will have financial implications, as many aspects of care for people with dementia is classified as social care and will be means tested. This is an important distinction between feasibility in the IP-SDM which focusses on availability of services and resources, and feasibility in our refined model. This directly fed into balancing the choices carefully.

Implementation and outcome are combined as participants did not distinguish between these two, and we highlight the reflective nature during this stage and iterative nature of decisions. The original IP-SDM does not clearly demonstrate that the outcomes of decisions have a substantial impact on subsequent decision-making and appears to sit outside of the decision-making process. However, in our refined model outcome is now an integral part of the process which feeds back into the entire process at each stage in subsequent decisions, as seen in Fig 1.

A key finding from this study is that despite social networks often considered important and supportive at the end-of-life [35], they are not always helpful or supportive when making decisions. Our model encourages reflection about the wishes, values and emotions of the person making decisions.

Our study has reinforced findings from previous work that making decisions as proxy for someone with dementia is a difficult and complex experience [11]. Many of the topics discussed in this paper and previous conversations described by participants did not always include the very end of life and lacked depth. Many struggle with the concept of their own mortality and avoided advance care planning (ACP) discussions [1, 17]. ACP is often seen as a tick box exercise [38] and our results suggest many may not understand how decisions may be beneficial to them later or what the implications of the decisions are. The model could help break down the complexity of decision making into stages and revisit decisions over time, congruent with meaningful ACP which should be conducted as a cyclical process [39].

## Strengths and limitations

This study is strengthened by the inclusion of both family carers and people living with dementia. People living with dementia are often excluded from EOLC research. The results are limited by the lack of inclusion of professionals. Professionals are a key member of the decision making team and are ultimately responsible for many significant decisions. However, many studies have been conducted with professionals about EOLC decision-making [13, 26], and the findings in the current study reflect these previous findings, further validating our model.

Our sample was predominantly white British. It is important to consider cultural factors which influence the decision-making process. People with dementia from black and minority ethnic backgrounds are less likely to receive early diagnosis, potentially making the window to have some of these discussions smaller [40].

Focussing on EOLC in the interviews may have missed the opportunity to explore how experiences, decisions, and discussions earlier in the trajectory influenced future decision-making behaviour. The interviews were conducted in the UK and although may transfer to other similar contexts in other countries there will be differences which limit the transferability of our findings to other countries health and care systems.

We used an existing conceptual model, the IP-SDM, overall, this was a good fit for our data and with some modifications has helped to break down the complexity of decision making in dementia EOLC.

## Implications for research, policy and practice

Many of the decisions participants discussed in this study were not necessarily specific to EOLC and we suggest this model may be applicable throughout the course of dementia, with some additional modifications. For example, decisions regarding ensuring everyday wellbeing of the person with dementia managing financial and household affairs. This model is the first step and will require further refinement, and testing including exploring suitability in the wider dementia context.

This model has already been useful and applied to support clinical practice. The model has directly informed the development of a decision aid to support family carers towards the end-of-life, and a second decision aid to support decision making for people with dementia in the context of COVID-19 [41], which has been implemented in UK National guidance on Covid-19 and dementia care [42] and used in clinical practice.

With more work to operationalise the model it may be further used, to provide a framework and communication tool for discussions with family carers and/or professionals, but also with the person with dementia providing a framework for how to initiate discussions and how to conduct meaningful ACP [43]. It may be helpful for those without a carer to use for themselves as a way of planning what they would like to happen when they lose capacity. Importantly, the model helps to understand where and how we can better align support with decision making on an individual level. Our study suggests there are several areas to target when supporting family carers with decision making. It is important to consider the emotional consequences of decision-making, and not underestimate the support needed with these decisions. Family carer should be reassured that decisions are not theirs alone to make and professionals will make significant decisions in the best interests of the individual.

Support should also cover the overall process of making decisions and not just focus on individual stages, for example support should enable carers to develop skills to manage decision making such as communication strategies and managing disagreement.

## Conclusions

Many carers and people with dementia struggle to discuss EOLC. Although the stages of the decision-making process appear inter-linked, our model breaks down and attempts to simplify the process while capturing the subtle nuances of decision making. Importantly we have highlighted that decision-making is not linear but rather a cyclical process.

## Acknowledgments

We would like to thank all the people living with dementia and family carers who took part in this study.

## Author Contributions

**Conceptualization:** Nathan Davies.

**Data curation:** Nathan Davies, Tanisha De Souza.

**Formal analysis:** Nathan Davies, Tanisha De Souza, Greta Rait, Jessica Meehan, Elizabeth L. Sampson.

**Funding acquisition:** Nathan Davies, Greta Rait, Elizabeth L. Sampson.

**Investigation:** Nathan Davies.

**Methodology:** Nathan Davies.

**Project administration:** Nathan Davies.

**Supervision:** Nathan Davies, Greta Rait, Elizabeth L. Sampson.

**Writing – original draft:** Nathan Davies.

**Writing – review & editing:** Nathan Davies, Tanisha De Souza, Greta Rait, Jessica Meehan, Elizabeth L. Sampson.

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
