## [Decision Letter · Decision Letter 0]

3 Mar 2021

PONE-D-20-36588

Making decisions towards the end of life about care for someone with dementia: Conceptualising a model of decision-making

PLOS ONE

Dear Dr. Davies,

Thank you for submitting your manuscript to PLOS ONE. After careful consideration, we feel that it has merit but does not fully meet PLOS ONE’s publication criteria as it currently stands. Therefore, we invite you to submit a revised version of the manuscript that addresses the points raised during the review process.

Please provide a clearer rationale for the need for a new model.  The paper would also benefit from much greater detail in the presentation of the results.

We look forward to receiving your revised manuscript.

Kind regards,

Rosemary Frey

Academic Editor

PLOS ONE

Journal Requirements:

Reviewers' comments:

Reviewer's Responses to Questions

**Comments to the Author**

1. Is the manuscript technically sound, and do the data support the conclusions?

Reviewer #1: No

Reviewer #2: Partly

2. Has the statistical analysis been performed appropriately and rigorously? 

Reviewer #1: N/A

Reviewer #2: N/A

3. Have the authors made all data underlying the findings in their manuscript fully available?

Reviewer #1: No

Reviewer #2: No

4. Is the manuscript presented in an intelligible fashion and written in standard English?

Reviewer #1: Yes

Reviewer #2: Yes

5. Review Comments to the Author

Reviewer #1: This manuscript presents a model of decision-making process for people with dementia. The main problem is that it does not clearly document the need for modification of an existing model. The information in one section (Lines 218 -298) does not justify this modification of the IP-SDM model. It should be more clearly explained why the modification is needed and the modification should be supported by quotes. Only after that, the modified model should be presented.

Minot points:

1. Line 88: Decision aid developed by Arcand is not mentioned.

2. Lines 249 and 262: Explain abbreviation

3. Line 260: Who is the respondent?

4. Line 618: Add a reference?

Reviewer #2: Thank you for the opportunity to review this interesting manuscript. Overall it is well written and raises some interesting points about the process of decision making in the dementia population. It is great to see people with dementia being included as this is rarely the case in dementia research.

However, I am not entirely convinced that a “new model of decision making” has been presented here. Instead the study has highlighted a “decision making process” in relation to people with dementia and their family. Defining what is meant by a decision making model along with a more comprehensive outline of what the new model looks like would help. Figure 1 appears to outline a process rather than a model.

Abstract

Small typo: Many are not ready to have conversations about end-of-life, and decision-making is left to their families and professionals when they no longer has (should this be have) capacity.

Introduction

Page 3 line 53-53: “People living with dementia have a right to be involved in decisions about their care and many countries have passed laws to this affect.” I suspect that this would only be the case in developed or resource rich countries and arguably in Western countries where autonomous decision making is prioritised. Please clarify this.

Page 3-4 Lines 66-69: Lasting power of attorney needs explaining in relation to the country this study was undertaken in. The reason for this is that the language used to describe legal proxy decision making differs internationally. For example, in some countries power of attorney needs to be “activated” before decisions can be made on the patient’s behalf i.e. the patient needs to be assessed as being unable to make their own decisions.

Page 4 lines 76-68: “Many people with dementia and family carers may not want to have discussions about EOLC.” This would benefit from a reference.

End of life care and end of life care conversations is used throughout – please define what is meant by EOLC and what might an EOLC conversation consists of.

Exclusion criteria – please explain what a Patient and Public Involvement (PPI) group is and their relationship with the study and who is on this group?

Procedure Page 7

Informed consent when participants have cognitive impairment is an important ethical issue to address. More detail on how this was achieved would be useful. For example, was an assessment of their cognition and capacity to consent made prior? How did you know that participants with dementia were well informed and had the capacity to participate in the study?

Were the two pilot interviews undertaken when piloting the interview schedule included in the final analysis? Were any changes made to the interview schedule as a result of the piloting?

Page 9 lines 199-201: “We presented the model and supporting data to our PPI group consisting of four former family carers of people with dementia to increase rigour of our model and findings.” How did this increase rigour of the findings and what was the feedback from this group? Did their feedback change anything in the model or the findings? Normally findings would be fed back to participants to improve rigour. Also, why were patients with dementia not included in this part of the process?

Findings

Contextual factors: At times when the authors referred to “participants” it is difficult to know whether they were family participants or patient participants. This is important when understanding the significance of the findings. Could you perhaps delineate more clearly between the two different participants? For example, on page 13 lines 2882-283 the authors state “Many participants talked of conflicts in their relationships with friends, neighbours and family members.” Whilst I appreciate the quote used to support the point being made is noted as a carer, when stating that “most” participants felt this ways it left me wondering if it was most carers or most patients or most overall.

The process: Page 16 lines 349-350: The authors state that “When sharing and exchanging information, there should be a consideration that many may be in denial?” Is that something that was picked up in the interviews?

Being “in denial” is arguably a useful coping mechanism for some people and may in fact be a reflection of people’s preference to not have these conversations. I am not entirely convinced that the quotes provided supports the argument of people being in denial rather it seems that either out of fear (“…a form of cowardice”) or preference (“…no need to keep going back and talking about it”) people choose to not enter into these conversations. Can more quotes be provided that would support this theme?

Loss of agency and control: provides a really interesting dyad conversation and raises a good point about loss of agency however I was left wondering if the person with dementia truly understood the questions being asked How did the interviewers ensure that the person with dementia understood the significance of the questions being asked?

Similarly the dyad example used in the section “Impact on family” – I was left again wondering if the question asking the patient to explain what they meant by being ‘looked after’ at the time when he had “run out of brain power’ was entirely understood by the participant. Their response is difficult to understand in relation to the question being asked.

Page 25 lines 529-530 The sentence “Importantly decision-making was not a linear process, decision making is a continuous process, iterative and live with decisions changed and updated throughout the course of providing care” – doesn’t make sense; is there a word missing?

Discussion

It is unclear how the “new decision-making model” differs from the IP-SDM and how it might have been used to inform the new model. What are the deficits in the IP-SDM model in relation to people with dementia and their carers? Could figure 1 be combined with the IP-SDM model so it is clear where the differences lie?

The discussion could be strengthened with more focus on the key findings and how these relate to what is already known in the literature about health and EOLC decision making in general and where possible in the dementia population.

Limitations – I suggest that the generalisability of the findings to other countries and healthcare systems is another limitation.

In addition, limiting the interview questions to EOLC may result in findings which are not reflective of the myriad of decision-making people have to make throughout the dementia trajectory. Indeed, these experiences may influence people’s future decision-making behaviours. This is particularly important to note given that the authors have suggested that the new model could be used across all decision making not just EOLC.

Implications for research, policy and practice: The authors suggest that the model will be useful in clinical practice however it is important to explain how the model be operationalised into clinical practice. I would suggest that more research is needed on how to operationalise and socialize a theory/model/process into practice. Transition of theory into practice is often seen as a major barrier in improving care “at the bedside”.

Figure 1: I think would benefit from more detail. For example, is this model an EOLC decision making process as indicated currently or it is an outline of the “new model” as highlighted in the discussion?

6. PLOS authors have the option to publish the peer review history of their article (what does this mean?). If published, this will include your full peer review and any attached files.

Reviewer #1: No

Reviewer #2: No

---

## [Author Response · Author response to Decision Letter 0]

30 Mar 2021

Dear Dr Frey and reviewers,

We would like to thank the editor and both reviewers for reviewing and providing such positive and constructive comments on our article. We have taken on board all your comments and revised our manuscript which we now feel is much stronger and clearer. 

We have provided detailed responses to each comment and marked it on the tracked changes version of our revised manuscript. 

Yours sincerely,

Dr Nathan Davies (on behalf of all authors)

Reviewer #1: This manuscript presents a model of decision-making process for people with dementia. The main problem is that it does not clearly document the need for modification of an existing model. The information in one section (Lines 218 -298) does not justify this modification of the IP-SDM model. It should be more clearly explained why the modification is needed and the modification should be supported by quotes. Only after that, the modified model should be presented.

Thank you for your comments on our paper. We have taken on board all your comments. It is important to note we have refined a model and adapted to a specific applied circumstance. We have added further justification for refining and adapting the IP-SDM model in the methods page 10 lines 207-214 and also in the discussion provided substantially more detail on the refined model, and how it differs from the IP-SDM. We provide quotes throughout the results which justifies the sections to the revised model presented in figure 1. The IP-SDM was a key part of our analysis to develop a refined model, our themes represent the model and as such we present this with our findings not after. 

Minot points:

1. Line 88: Decision aid developed by Arcand is not mentioned.

We have cited a systematic review of decision aids, which covers a range of different decision aids published.

2. Lines 249 and 262: Explain abbreviation

We have added an explanation, now lines 284 and 298. 

3. Line 260: Who is the respondent?

We have clarified this throughout the results where it is not clear. 

4. Line 618: Add a reference?

This is a statement about the results in this study and other published studies, however we have added an example and refer back to decisions stated at the start of the results section. This is now line 728 – 733.

Reviewer #2: Thank you for the opportunity to review this interesting manuscript. Overall it is well written and raises some interesting points about the process of decision making in the dementia population. It is great to see people with dementia being included as this is rarely the case in dementia research.

Thank you for your positive comments and your points raised which we feel have been very helpful to strengthen out paper. 

However, I am not entirely convinced that a “new model of decision making” has been presented here. Instead the study has highlighted a “decision making process” in relation to people with dementia and their family. Defining what is meant by a decision making model along with a more comprehensive outline of what the new model looks like would help. Figure 1 appears to outline a process rather than a model.

We have refined an existing model and adapted it to the specific circumstances of dementia and of life care. The IP-SDM model of decision making is well established as a model in the literature. 

We agree this consists of the decision-making process, however this also includes a consideration of contextual factors which influence the decision-making process, conceptualising a model of how decisions are made. 

We have used the following explanation from Nilsen 2015 of models “A model typically involves a deliberate simplification of a phenomenon or a specific aspect of a phenomenon. Models need not be completely accurate representations of reality to have value. Models are closely related to theory and the difference between a theory and a model is not always clear. Models can be described as theories with a more narrowly defined scope of explanation; a model is descriptive, whereas a theory is explanatory as well as descriptive.” We have summarized this on page 9 lines 187-190.

We have revised figure 1 to better reflect this is more than a process and discuss this in the discussion. 

Abstract

Small typo: Many are not ready to have conversations about end-of-life, and decision-making is left to their families and professionals when they no longer has (should this be have) capacity.

This has been corrected. 

Introduction

Page 3 line 53-53: “People living with dementia have a right to be involved in decisions about their care and many countries have passed laws to this affect.” I suspect that this would only be the case in developed or resource rich countries and arguably in Western countries where autonomous decision making is prioritised. Please clarify this.

We have added clarity to this statement now line 53-55.

Page 3-4 Lines 66-69: Lasting power of attorney needs explaining in relation to the country this study was undertaken in. The reason for this is that the language used to describe legal proxy decision making differs internationally. For example, in some countries power of attorney needs to be “activated” before decisions can be made on the patient’s behalf i.e. the patient needs to be assessed as being unable to make their own decisions.

We have added a point to say this is UK based and this will vary internationally see lines 69-71. 

Page 4 lines 76-68: “Many people with dementia and family carers may not want to have discussions about EOLC.” This would benefit from a reference.

We have added a reference – this is now line 86.

End of life care and end of life care conversations is used throughout – please define what is meant by EOLC and what might an EOLC conversation consists of.

We have added on page 4 lines 77-80 a short description of what EOLC conversations may include. 

Definitions of eolc in dementia are notoriously difficult and there is no agreed definition in this population. We have added a statement for this and have been careful in our title and throughout the paper by using the phrasing ‘towards’ the end of life so we do not place a set time period on this. See lines 80-83.

Exclusion criteria – please explain what a Patient and Public Involvement (PPI) group is and their relationship with the study and who is on this group?

We have added a statement on this to the design section pages 6 -7, lines 134-136.

Procedure Page 7

Informed consent when participants have cognitive impairment is an important ethical issue to address. More detail on how this was achieved would be useful. For example, was an assessment of their cognition and capacity to consent made prior? How did you know that participants with dementia were well informed and had the capacity to participate in the study?

We conducted a short assessment using the principles of the UK Mental Capacity Act. We have added this detail to page 8 lines 165-169.

Were the two pilot interviews undertaken when piloting the interview schedule included in the final analysis? Were any changes made to the interview schedule as a result of the piloting?

They were included in the final analysis. Minor amendments were made to the topic guide for phrasing based on pilot interviews and iteratively refined throughout the study. We have added a sentence on both these points on page 8 lines 176-178.

Page 9 lines 199-201: “We presented the model and supporting data to our PPI group consisting of four former family carers of people with dementia to increase rigour of our model and findings.” How did this increase rigour of the findings and what was the feedback from this group? Did their feedback change anything in the model or the findings? Normally findings would be fed back to participants to improve rigour. Also, why were patients with dementia not included in this part of the process?

The role of a PPI group is to provide expertise from their lived experiences as a member of the population you are researching. Unfortunately, we were not able to recruit someone with dementia for this advisory group, but we had a breadth of expertise form four former carers. Their input allowed us to share with them our analysis and model allowing them to challenge our ideas, hence improving rigour by sharing ideas and increasing discussions, crucial in qualitative research. We have cited a reference for this. The feedback did not overall change the model but provided us with reassurance this matched their experiences and provide some further context outside our findings. 

We have added details of the ppi group to the design section (see previous comment) and we have added a sentence on page 11 lines 234-236.

Findings

Contextual factors: At times when the authors referred to “participants” it is difficult to know whether they were family participants or patient participants. This is important when understanding the significance of the findings. Could you perhaps delineate more clearly between the two different participants? For example, on page 13 lines 2882-283 the authors state “Many participants talked of conflicts in their relationships with friends, neighbours and family members.” Whilst I appreciate the quote used to support the point being made is noted as a carer, when stating that “most” participants felt this ways it left me wondering if it was most carers or most patients or most overall.

We have added clarity throughout. 

The process: Page 16 lines 349-350: The authors state that “When sharing and exchanging information, there should be a consideration that many may be in denial?” Is that something that was picked up in the interviews? Being “in denial” is arguably a useful coping mechanism for some people and may in fact be a reflection of people’s preference to not have these conversations. I am not entirely convinced that the quotes provided supports the argument of people being in denial rather it seems that either out of fear (“…a form of cowardice”) or preference (“…no need to keep going back and talking about it”) people choose to not enter into these conversations. Can more quotes be provided that would support this theme?

Thank you for this interesting point. We have added ‘may be in fear or in denial’, on page 18 line 390 and changed the title of the subtheme to ‘Questioning applicability to ‘us’ as an expression of denial, fear and avoidance‘. We also remark on how denial is implicitly expressed and therefore may not be obvious but is an interpretation. Finally, we have added two quotes to express denial pages 19-20 lines 424-432.

Loss of agency and control: provides a really interesting dyad conversation and raises a good point about loss of agency however I was left wondering if the person with dementia truly understood the questions being asked How did the interviewers ensure that the person with dementia understood the significance of the questions being asked?

The interviewer felt the person did understand. In any occurrences where they did not seem to understand, questions were repeated in a different way and in some occasions if interviewed as a dyad the carer also phrased the question in a way they thought the person would understand. However, a lack of understanding was rare. 

Similarly the dyad example used in the section “Impact on family” – I was left again wondering if the question asking the patient to explain what they meant by being ‘looked after’ at the time when he had “run out of brain power’ was entirely understood by the participant. Their response is difficult to understand in relation to the question being asked.

The point you raise is important and interesting and this is how we have interpreted it that their response reflects they do not want to discuss or accept their decline as we discuss, instead they focus on what they can do and why they don’t need to be looked after. We have added a point to clarify on page 24 line 534. The participants had mild dementia and were still able to understand the questions and discussions. 

Page 25 lines 529-530 The sentence “Importantly decision-making was not a linear process, decision making is a continuous process, iterative and live with decisions changed and updated throughout the course of providing care” – doesn’t make sense; is there a word missing?

We have clarified this sentence now page 27 line 587-588.

Discussion

It is unclear how the “new decision-making model” differs from the IP-SDM and how it might have been used to inform the new model. What are the deficits in the IP-SDM model in relation to people with dementia and their carers? Could figure 1 be combined with the IP-SDM model so it is clear where the differences lie?

We have substantially changed our discussion adding much more detail regarding the revised model and updated figure 1. We have also reordered our discussion to make this clearer. 

Our findings were mapped onto the model as detailed in the methods section, page 10-11, lines 216-236, following other similar published studies. It acted as a foundation model which we revised to fit the dementia eolc context as stated on page 27 lines 596-600.

The discussion could be strengthened with more focus on the key findings and how these relate to what is already known in the literature about health and EOLC decision making in general and where possible in the dementia population.

We are mindful of the length of the manuscript for readers so have tried to minimize the addition of new content, however we have further emphasized the contribution to knowledge and referenced further work in this field. We have added detail throughout the discussion including a reference on grief (page 30 line 673) and discussions about eolc on lines 661-667, we have highlighted disagreement and conflct on 662-665, responsibility 656-658, and inclusion of people with dementia in decision making linked to capacity pages 29-30 lines 649-651.

Limitations – I suggest that the generalisability of the findings to other countries and healthcare systems is another limitation.

We have added a statement discussing the transferability as opposed to generalisability as this is a qualitative study we are not looking to generalise, to the limitations section see page 32-33, lines 719-721.

In addition, limiting the interview questions to EOLC may result in findings which are not reflective of the myriad of decision-making people have to make throughout the dementia trajectory. Indeed, these experiences may influence people’s future decision-making behaviours. This is particularly important to note given that the authors have suggested that the new model could be used across all decision making not just EOLC.

We have added this to page 32 lines 717-719.

Implications for research, policy and practice: The authors suggest that the model will be useful in clinical practice however it is important to explain how the model be operationalised into clinical practice. I would suggest that more research is needed on how to operationalise and socialize a theory/model/process into practice. Transition of theory into practice is often seen as a major barrier in improving care “at the bedside”.

The model has already been used to inform two decision aids including one which is being used in clinical practice during covid-19 and cited in national guidance in the UK and a second which is currently being trialed in a study. We have amended this section to make this clear. However, we have also added a sentence (page 33 line 741) to recommend further work to operationalize the model for any further practical application regarding our other recommendations for supporting discussions and communication.

Figure 1: I think would benefit from more detail. For example, is this model an EOLC decision making process as indicated currently or it is an outline of the “new model” as highlighted in the discussion?

We have added more detail in the discussion and clarified the whole refined model as per earlier comments. We have also updated fig 1.

---

## [Decision Letter · Decision Letter 1]

11 May 2021

PONE-D-20-36588R1

Developing an applied model for making decisions towards the end of life about care for someone with dementia

PLOS ONE

Dear Dr. Davies,

Thank you for submitting your manuscript to PLOS ONE. After careful consideration, we feel that it has merit but does not fully meet PLOS ONE’s publication criteria as it currently stands. Therefore, we invite you to submit a revised version of the manuscript that addresses the points raised during the review process.

Please respond to the literature review queries raised by reviewer 1.

We look forward to receiving your revised manuscript.

Kind regards,

Rosemary Frey

Academic Editor

PLOS ONE

Journal Requirements:

Reviewers' comments:

Reviewer's Responses to Questions

**Comments to the Author**

1. If the authors have adequately addressed your comments raised in a previous round of review and you feel that this manuscript is now acceptable for publication, you may indicate that here to bypass the “Comments to the Author” section, enter your conflict of interest statement in the “Confidential to Editor” section, and submit your "Accept" recommendation.

Reviewer #1: (No Response)

Reviewer #2: All comments have been addressed

2. Is the manuscript technically sound, and do the data support the conclusions?

Reviewer #1: Yes

Reviewer #2: Yes

3. Has the statistical analysis been performed appropriately and rigorously? 

Reviewer #1: N/A

Reviewer #2: N/A

4. Have the authors made all data underlying the findings in their manuscript fully available?

Reviewer #1: Yes

Reviewer #2: Yes

5. Is the manuscript presented in an intelligible fashion and written in standard English?

Reviewer #1: Yes

Reviewer #2: Yes

6. Review Comments to the Author

Reviewer #1: The authors state that "We have cited systematic review of decision aids, which covers a range of different decision aids

published." However, at least one of these aids, published by Arcand, was not included. What were the creteria for selection of published decision aids and how many more were not included?

Reviewer #2: (No Response)

7. PLOS authors have the option to publish the peer review history of their article (what does this mean?). If published, this will include your full peer review and any attached files.

Reviewer #1: No

Reviewer #2: No

---

## [Author Response · Author response to Decision Letter 1]

11 May 2021

Dear Dr Frey and reviewers,

We would like to thank the editor and both reviewers for reviewing and checking our revision. 

We have provided detailed response to reviewer 1 comment below.

Yours sincerely,

Dr Nathan Davies (on behalf of all authors)

Reviewer 1

The authors state that "We have cited systematic review of decision aids, which covers a range of different decision aids published." However, at least one of these aids, published by Arcand, was not included. What were the creteria for selection of published decision aids and how many more were not included?

This was not included in the published systematic review as it is an information resource or booklet on comfort care, it is not described as a decision aid in any of the papers that were identified as part of that search, and does not contain the features of a decision aid. This was an exclusion criteria in the published systematic review. However, we agree it is a very important resource to support decisions which has been used in multiple studies and evaluated, we have added the Arcand paper to the introduction of the current paper see line 97.

---

## [Decision Letter · Decision Letter 2]

17 May 2021

Developing an applied model for making decisions towards the end of life about care for someone with dementia

PONE-D-20-36588R2

Dear Dr. Davies,

We’re pleased to inform you that your manuscript has been judged scientifically suitable for publication and will be formally accepted for publication once it meets all outstanding technical requirements.

Kind regards,

Rosemary Frey

Academic Editor

PLOS ONE

Additional Editor Comments (optional):

Reviewers' comments:

Reviewer's Responses to Questions

**Comments to the Author**

1. If the authors have adequately addressed your comments raised in a previous round of review and you feel that this manuscript is now acceptable for publication, you may indicate that here to bypass the “Comments to the Author” section, enter your conflict of interest statement in the “Confidential to Editor” section, and submit your "Accept" recommendation.

Reviewer #1: All comments have been addressed

2. Is the manuscript technically sound, and do the data support the conclusions?

Reviewer #1: Yes

3. Has the statistical analysis been performed appropriately and rigorously? 

Reviewer #1: N/A

4. Have the authors made all data underlying the findings in their manuscript fully available?

Reviewer #1: No

5. Is the manuscript presented in an intelligible fashion and written in standard English?

Reviewer #1: Yes

6. Review Comments to the Author

Reviewer #1: (No Response)

7. PLOS authors have the option to publish the peer review history of their article (what does this mean?). If published, this will include your full peer review and any attached files.

Reviewer #1: No